# Glucoraphanin Increases Intracellular Hydrogen Sulfide (H_2_S) Levels and Stimulates Osteogenic Differentiation in Human Mesenchymal Stromal Cell

**DOI:** 10.3390/nu14030435

**Published:** 2022-01-19

**Authors:** Laura Gambari, Marli Barone, Emanuela Amore, Brunella Grigolo, Giuseppe Filardo, Renato Iori, Valentina Citi, Vincenzo Calderone, Francesco Grassi

**Affiliations:** 1Laboratorio RAMSES, IRCCS Istituto Ortopedico Rizzoli, Via di Barbiano 1/10, 40136 Bologna, Italy; laura.gambari@ior.it (L.G.); marlibarone@gmail.com (M.B.); Manuamore@live.com (E.A.); brunella.grigolo@ior.it (B.G.); 2Applied and Translational Research Center (ATRc), IRCCS Istituto Ortopedico Rizzoli, Via di Barbiano 1/10, 40136 Bologna, Italy; giuseppe.filardo@ior.it; 3Research and Innovation Centre, Department of Food Quality and Nutrition, Fondazione Edmund Mach, 38098 San Michele all’Adige, Italy; renato.iori48@gmail.com; 4Department of Pharmacy, University of Pisa, Via Bonanno 6, 56126 Pisa, Italy; valentina.citi@unipi.it (V.C.); vincenzo.calderone@unipi.it (V.C.)

**Keywords:** glucoraphanin, osteoporosis, osteoblast, hydrogen sulfide, nutritional supplements, mesenchymal stromal cells

## Abstract

Osteopenia and osteoporosis are among the most prevalent consequences of ageing, urging the promotion of healthy nutritional habits as a tool in preventing bone fractures. Glucosinolates (GLSs) are organosulfur compounds considered relatively inert precursors of reactive derivatives isothiocyanates (ITCs). Recent evidence suggests that GLSs may exert biological properties based on their capacity to release hydrogen sulfide (H_2_S). H_2_S-donors are known to exert anabolic function on bone cells. Here, we investigated whether a GLS, glucoraphanin (GRA) obtained from Tuscan black kale, promotes osteogenesis in human mesenchymal stromal cells (hMSCs). H_2_S release in buffer and intracellular H_2_S levels were detected by amperometric measurements and fluorimetric/cytofluorimetric analyses, respectively. Alizarin red staining assay and real-time PCR were performed to evaluate mineral apposition and mRNA expression of osteogenic genes. Using an in vitro cell culture model, our data demonstrate a sulforaphane (SFN)-independent osteogenic stimulation of GRA in hMSCs, at least partially attributable to H_2_S release. In particular, GRA upregulated the expression of osteogenic genes and enhanced mineral apposition while increasing intracellular concentrations of H_2_S. Overall, this study suggests the feasibility of using cruciferous derivatives as natural alternatives to chemical H_2_S-donors as adjuvant therapies in the treatment of bone-wasting diseases.

## 1. Introduction

Dietary habits are an important determinant of bone health [1]. In particular, certain micronutrients contained in fruit and vegetables contribute to delaying bone fragility in ageing and decreasing the incidence of bone fractures [2,3,4,5,6]. 

Glucosinolates (GLSs) are a group of organosulfur compounds of natural origin, commonly regarded as the precursors of isothiocyanates (ITCs). In plants, GLSs are the substrate of the b-thioglucosidase enzyme myrosinase, which triggers the hydrolytic cleavage of the GLS molecule, releasing glucose and originating an unstable aglycone which, at physiological pH, is mostly rearranged to highly reactive ITC [7,8]. In mammals, the same catabolic breakdown of GLSs is carried out only by microbial thioglucosidases of the gut microbiota, given that myrosinase is not expressed in mammalian cells [9]. 

Plants belonging to the family of Brassicaceae, also known as cruciferous vegetables, are the most abundant source of naturally occurring GLS. Diets rich in Brassicaceae were associated with several health benefits, such as the maintenance of cardiovascular and metabolic health [10], reduced risk of cancer [11], and protection against neurologic disorders [12]. Notably, several clinical trials have documented the effect of GLS on human health [13]. Glucoraphanin (4-methylsulphinylbutyl glucosinolate (GRA)), a chemically stable GLS, is abundant in certain cruciferous vegetables, including broccoli, cabbages, cauliflowers, brussels sprouts, rocket, kohlrabi, radish [14]. Moreover, the seeds of Tuscan black kale were found to be an abundant and reliable source of GRA [15]. Upon hydrolysis, GRA is converted to sulforaphane (4-methylsulphinylbutyl isothiocyanate (SFN)), a potent inducer of the KEAP1/NRF2/ARE pathway, leading to the activation of a potent antioxidant and detoxifying response in cells [16,17] as well as anti-inflammatory [18,19] and antiapoptotic effects [20]. 

GRA is predominantly considered a biologically inert molecule [21], and its biological properties have been mostly attributed to SFN. However, in a recent work, Lucarini et al. found that GRA releases hydrogen sulfide (H_2_S) in a cysteine-dependent fashion, in a manner very similar to its cognate ITC, SFN [22,23], suggesting that GRA may be endowed with biological activity based on its H_2_S-releasing properties [24].

H_2_S, the ubiquitous gasotransmitter first characterized in human tissues in 1996 [25], is produced endogenously via enzymatic and nonenzymatic pathways in mammalian cells [26]. If maintained within a physiological or slightly supraphysiological level, H_2_S provides a number of health benefits by improving cardiometabolic disorders [27,28], relieving pain [29,30], attenuating ischemia–reperfusion injury [31], and increasing insulin sensitivity [32]. Moreover, H_2_S is critically involved in the lifespan extension provided by caloric restriction [33,34]. In bone, H_2_S plays an anabolic role by promoting mesenchymal stromal cells (MSCs) osteogenic differentiation by stimulating the WNT pathway and regulating MSCs’calcium intake [35,36]. In addition, H_2_S acts by inhibiting osteoclast differentiation in vitro [37,38]. As a result, the pharmacological administration of H_2_S-donors promotes bone formation in vivo [35,36] and mitigates the bone-wasting effects of estrogen deficiency and corticosteroids [35,39]. 

The biological activity of organosulfur molecules typical of Brassicaceae has been increasingly linked to their H_2_S-releasing capacity [27,40]. Indeed, not only a broad number of ITCs were shown to behave as H_2_S-releasing molecules in solution [41], but the pain-relieving effect of the ITC SFN has also been directly attributed to H_2_S release using an H_2_S-binding molecule, haemoglobin, in a model of chemotherapy-induced neuropathy [24]. Whether GLS exerts biological properties mediated by H_2_S release in bone cells remains to be elucidated. 

This study aimed to investigate the H_2_S-releasing activity of GRA and to assess its effect on osteogenic differentiation in a model of in vitro cell culture of human MSCs.

## 2. Materials and Methods

### 2.1. Extraction, Isolation and Characterization of GRA from Seeds

GRA was purified from *Brassica oleracea* L. var. acephala sabellica ripe seeds supplied by SUBASEEDS (Longiano, Italy). Seeds were first ground to a fine powder and defatted in hexane. After removing the solvent, the defatted meal was treated with boiling 70% ethanol to produce a quick deactivation of endogenous myrosinase and to extract the intact GLS. The isolation of GRA from the extract was carried out by one-step anion-exchange chromatography, as previously described [42]. The purity was further enhanced by gel-filtration, which was performed using an XK 26/100 column packed with Sephadex G10 chromatography media (Amersham Biosciences, Milano, Italy), connected to an FPLC System (Pharmacia, Milano, Italy). The mobile phase was water at a 2.0 mL/min flow rate, and the eluate absorbance was monitored at 254 nm. Fractions were assayed by high-performance liquid chromatography (HPLC) analysis of the desulpho-derivative according to the ISO 9167-1 method, and those containing the GRA (>95%) were collected and freeze-dried. GRA was characterized by ^1^H and ^13^C NMR spectrometry, and the absolute purity estimated by HPLC was close to 95%.

### 2.2. Determination of H_2_S Release in Buffer by Amperometric Assay

The evaluation of H_2_S-release by GRA has been carried out by an amperometric approach through the Apollo-4000 free radical analyzer (WPI) detector and H_2_S-selective mini-electrodes, as previously described [43]. Briefly, “PBS buffer 10×” was prepared and diluted to PBS 1×, immediately before the use. The H_2_S-selective mini-electrode was equilibrated in 10 mL of PBS 1× (composition of PBS 10×: NaH_2_PO_4_·H_2_O 1.28 g, Na_2_HPO_4_·12H_2_O 5.97 g, NaCl 43.88 g in 500 mL H_2_O), until the recovery of a stable baseline. Then, L-Cysteine (final concentration 4 mM) was added 10 min before adding GRA 1 mM, and the generation of H_2_S was observed for 20 min. Notably, L-Cysteine alone did not produce any amperometric response. A calibration curve, obtained by using sodium hydrosulfide (NaHS; 1-3-5-10 μM), a high releasing H_2_S donor, was performed to obtain concentrations of H_2_S from amperometric currents measurement (recorded in pA).

### 2.3. Patients

The study has been approved by the Institutional Ethics Committee (CE 0011810/2017) and conducted according to national and international legislations, to principles of the ICH-GCP and to Helsinki Declaration (Fortaleza, October 2013). Cells were isolated from bone chips of tibial plateau obtained by patients undergoing surgical knee replacement. All surgical procedures and the harvesting of human tissues were performed at the Rizzoli Orthopedic Hospital after having obtained patients’ informed consent. Samples were obtained from patients of both genders, aged 55–85. A total of six women (mean age 74, range 58–84) and three men (mean 65, range 59–76) were enrolled. Obese patients (BMI > 30), patients affected by major chronic diseases affecting bone metabolism (diabetes, rheumatic diseases, infectious diseases, tumours, psychiatric diseases), and patients undergoing pharmacological therapies with steroids were excluded from the study.

### 2.4. Cell Isolation and Culture

hMSCs were isolated using a mechanical and a Ficoll density gradient isolation protocol [44]. Cells were cultured in α-MEM 15% FBS at 37 °C, 5% CO_2_ and 95% O_2_ and the medium was replaced twice per week; they were expanded until passage 2 when they were harvested with trypsin/EDTA solution 0.25% (Biochrom, Berlin, Germany) and seeded for osteogenic cultures (at passage 3).

### 2.5. Measurement of H_2_S in hMSCs Cell Culture

A total of 3 hMSCs donors were employed in these set of experiments. Intracellular H_2_S levels were detected by fluorimetric analysis, immunofluorescence, and flow cytometry assays using WSP-5 (Cayman Chemical, Ann Arbor, MI, USA), a fluorescent probe based on nucleophilic substitution–cyclization [45]. NaHS (Thermo Fisher Scientific, NJ, USA) was used as a positive control. 

For the fluorimetric analysis, cells were seeded at 10^4^ cells/well on 96-well plates in α-MEM 15% FBS at 37 °C and with 5% CO_2_. Cells were washed with D-PBS and further incubated with D-PBS added with 50 μM WSP-5 and 100 μM Hexadecyltri-methylammonium bromide (CTAB, Sigma Aldrich, St. Louis, MO, USA) at 37 °C for 30 min. After removing excess probe by washing with D-PBS, cells were treated with D-PBS and 3.3–100 μM GRA and NaHS. Only in this set of experiments, we used NaHS at 25 μM as a positive control to guarantee detection of progressive increase in intracellular H_2_S levels and avoid an instant plateau of H_2_S concentration, which could be obtained by using the standard reference concentration 200 μM NaHS, given the nature of NaHS as a fast-releasing H_2_S donor. Control wells were left untreated to detect basal intracellular levels of H_2_S. A time-course measurement of intracellular H_2_S was performed by fluorimetric analysis at 1-5-10-15-20-30 min and 1–2 h with a Spectra-Max Gemini fluorimeter (Molecular Probes) at a 525 nm emission wavelength. 

For the immunofluorescence assays, hMSCs were cultured at 30,000 cells/wells in chamber slides with α-MEM 15% FBS at 37 °C and with 5% CO_2_. The day after the seeding, 100 μM GRA or 200 μM NaHS was added to the media and incubated for 3 h. Two wells were left untreated to obtain the control without WSP-5 (CTRL, control of autofluorescence) and the control with WSP-5 (CTRL-, control of the basal intracellular levels of H_2_S). Cell culture medium was then removed, and cells were washed with D-PBS. Afterwards, the cells were incubated with D-PBS added with 50 μM WSP-5 and 100 μM CTAB at 37 °C for 30 min. After removing excess probe by washing with D-PBS, cells were treated with D-PBS and 100 μM GRA or 200 μM NaHS and 100 μM CTAB. After 1 h of incubation, cells were washed with D-PBS, fixed in PFA 4% for 20 min, washed with D-PBS, and finally mounted with antifading containing DAPI (ProLong Diamond antifade mountant with Dapi). Images were captured by analyzing FITC fluorescence with Nikon Instruments Europe BV (Amstelveen, The Netherlands).

For the flow cytometry assays, we performed the experiments as previously reported [46]. Briefly, after harvesting with trypsin/EDTA solution 0.25% (Biochrom) and washing in PBS 1×, 3 × 10^5^ fresh hMSCs were incubated for 30 min at 37 °C in buffer BS (composition: HEPES 20 mM, NaCl 120 mM, KCl 2 mM, CaCl_2_ × 2H_2_O 2 mM, MgCl_2_ × 6H_2_O 1 mM, glucose 5 mM) with 50 μM WSP-5 and 100 μM CTAB and washed in PBS 1×. Afterwards, cells were incubated with buffer BS with 100 μM CTAB and 100 μM GRA or 200 μM NaHS and assessed at 1 h after stimulation by flow cytometry analysis performed on FACS canto II (BD bioscience, San Jose, CA, USA): 224 voltage (FITCH); Threshold 33303.

### 2.6. Osteogenic Differentiation and Alizarin Red Staining

A total of six donors of hMSCs were employed in these sets of experiments. hMSCs from each donor were seeded at passage 3 at 5 × 10^4^ cells/cm^2^ in 12 wells plate in α-MEM 15% FBS and cultured for 14–21 days in osteogenic medium (α-MEM 20% FBS supplemented with 100 nM dexamethasone, 100 µM ascorbic acid, and 10 mM β-glycerophosphate) with or without treatment with 3.3, 10, 33, and 100 μM GRA, each condition was seeded in duplicates. Medium and stimuli were replaced twice per week. On day14 and day 21, Alizarin Red S (AR-S) (Sigma Aldrich) staining was performed to assess the presence and extent of mineralization. Briefly, cells were stained with 40 mM AR-S for 20 min after being fixed for 15 min at RT in formaldehyde (Kaltek, Padova, Italy) 10% phosphate buffered saline (PBS) and washed twice with PBS, as detailed elsewhere. A spectrophotometric analysis with TECAN Infinite^®^ 200 PRO (Tecan Italia S.r.l., Cernusco Sul Naviglio, Italy) was performed to quantify the mineral apposition, as described in our previous study [44]. In particular, 177 readings were recorded from each well and averaged. Control cells at day 0 showed an average of 0.2, which we considered as the background, negative control of mineralization. A value of 0.3 corresponds to the baseline mineralization, the lowest level of mineralization detected by this method. Intermediate levels of mineralization are between 0.3 and 0.9, and high levels of mineralization are above 0.9 (where 100% of the 177 readings were above the baseline of mineralization). Nikon Instruments Europe BV (Amstelveen, The Netherlands) was used to obtain photos at 100× magnification. 

### 2.7. Quantification of Gene Expression by Real-Time PCR

A total of 6 hMSCs donors were employed to assess gene expression during osteogenic stimulation. hMSCs were seeded at passage 3 and cultured under osteogenic stimulation in the presence or absence of 3.3, 10, 33, and 100 μM GRA (as detailed above, each condition was seeded in duplicates). At the end of culture, cells were lysed using 1 mL of RNA pure solution (Euroclone, Milan, Italy) before performing the chloroform–phenol-ethanol extraction protocol and the purification from genomic DNA by treatment with DNase I (DNA-free Kit, Ambion, Austin, TX, USA)*,* according to manufacturer instructions. cDNA synthesis was performed by using SuperScript™ VILO™ cDNA Synthesis Kit (Invitrogen) on 2720 Thermal cycler (Applied Biosystem, Life Technologies) at 25 °C for 10 min, 42 °C for 60 min, 85 °C for 5 min, and 4 °C for 30 min. mRNA expression was assessed by real-time polymerase chain reaction (PCR) analysis using the SYBR Premix Ex Taq (TaKaRa Biomedicals, Tokyo, Japan). Primers were purchased from Life Technologies Italia (primers sequences are reported in Table 1). The real-time PCR analyses were run on LightCycler Instrument (Roche) as follows: one cycle at 95 °C for 10 s, 45 cycles at 60 °C for 20 s, and at 95 °C for 5 s. Standard melting curve analyses were performed at 95 °C for 10 s, 65 °C for 15 s, and 95 °C in one-degree increments for confirming the specificity of the PCR products. PCR products were relatively quantified with the comparative C_T_ method, comparing to the housekeeping mRNA expression of glyceraldehyde-3 phosphate dehydrogenase (GAPDH). 

### 2.8. Statistical Analyses

GraphPad Prism 7 (La Jolla, CA, USA) was used for statistical analyses. Before each test, the presence of outliers was checked by the ROUT (Q = 1%) test. Outliers were removed from each data set when present. D’Agostino & Pearson normality test was performed to analyze the normality of our data. Data sets that showed an ideal Gaussian distribution were assayed with one-way ANOVA and Dunnett’s multiple comparison tests or two-way ANOVA for repeated measures by two factors (time and treatment) and Dunnett’s multiple comparison test vs. control cells at each time point. Data set which did not show an ideal Gaussian distribution were assayed with Wilcoxon signed-rank test. Wilcoxon signed-rank test vs. reference value 0.2 (the baseline value of control at day 0, corresponding to the absence of mineralization) was performed when analyzing the data set of alizarin red staining comparing day 14 and day 21 to day 0. Significance was attributed when *p* < 0.0001 (****), *p* < 0.001 (***), *p* < 0.01 (**) and *p* < 0.05 (*).

## 3. Results

### 3.1. GRA Increases Intracellular H_2_S Levels in hMSCs

L-cysteine-dependent release of H_2_S is a common feature of SFN [23] and other ITCs [41]. Recent work reported that GRA behaves as an H_2_S-donor independent of the hydrolysis to SFN triggered by plant or microbial myrosinase [24]. Prompted by this observation, we assessed H_2_S levels via amperometric measurement in an aqueous solution in the presence of 1 mM GRA. Confirming the previous report, GRA released H_2_S only in the presence of L-cysteine, shown in Figure 1a. To investigate whether the stimulation with GRA results in increased H_2_S levels, even in the intracellular microenvironment, we exposed hMSCs to different concentrations of GRA and performed fluorimetric measurement of H_2_S using a specific fluorescent probe. Figure 1b shows that stimulation with GRA induced a dose-dependent increase of intracellular H_2_S levels within the timeframe chosen for this experiment (2 h). Interestingly, H_2_S levels appeared to slowly increase over time, suggesting that GRA may act as a slow-releasing H_2_S donor. The fast H_2_S-donor NaHS added to the cell culture at the concentration of 25 μM was used as positive control and showed a higher and faster H_2_S intracellular increase.

To further confirm that GRA increases H_2_S levels in hMSCs, and to exclude that fluorescent signal may derive, at least in part, from a leakage of the selective probe WSP-5 outside the cell membrane, we repeated the experiment by selecting the high concentration of 100 μM GRA, detaching the cell by trypsin treatment, and detecting intracellular H_2_S-specific fluorescence by flow cytometry. As shown in Figure 2a,b, GRA-treated cells showed a 1.5-fold increase in median fluorescence intensity compared with control, unstimulated cells, thereby confirming that intracellular H_2_S is increased upon GRA stimulation. As expected, NaHS used as positive control showed a much stronger increase in cell fluorescence signal. Finally, pictures of hMSCs treated with WSP-5 and the nuclear probe DAPI are shown in Figure 2c. Once again, GRA-treated cells showed a stronger fluorescence signal than unstimulated cells.

### 3.2. GRA Increases Mineralization in Osteogenic Cultures of hMSCs

The effects of long-term stimulation with GRA were tested in a model of osteogenic differentiation of hMSCs for up to 21 days. Given the heterogeneity in response to osteogenic stimulation of in vitro culture of primary hMSCs, we first analyzed the kinetics of production of the mineral matrix of our hMSCs population. All six donors analyzed showed early-intermediate levels of mineralization at day 14, while they all reached the highest levels of differentiation allowed by this in vitro model of differentiation (values > 0.9) at day 21 (Figure 3b). Figure 3a shows a representative picture of cell culture mineralization at each time point.

When hMSCs were stimulated with different concentrations of GRA (3.3, 10, 33, and 100 μM), we observed an increase in the mineral matrix apposition (Figure 4a,b). The significant increase in mineralization by GRA stimulation was evident at the early time point day 14, where the majority of unstimulated cells were still in the exponential phase and expressed different grades of mineral deposition. As reported in Figure 4a, the highest relative increase in mineralization was observed at the highest dose of 100 μM (*p* < 0.01); however, GRA induced a similar increase in mineralization even at the low dose of 3.3 μM; (*p* < 0.05). By day 21, when the mineralization had levelled off and reached the highest levels even in control samples, GRA-treated samples still showed a slightly higher density of mineralized extracellular matrix compared to unstimulated samples (Figure 4b). However, the magnitude of the increase was lower, and none of the concentrations achieved statistically significant differences compared to controls. 

### 3.3. GRA Stimulates Expression of Osteogenic Markers in hMSCs

The effect of GRA on osteogenesis was also assessed on the mRNA expression of genes involved in osteoblastic differentiation of hMSCs. Figure 5 summarizes data from samples collected on days 0,14, and 21. Interestingly, GRA stimulation affected the expression of BSP, a member of the SIBLING (small, integrin-binding ligand N-linked glycoprotein) family of a protein implicated in the initiation of hydroxyapatite crystal formation in the bone matrix. At the same time, the levels of BSP increased over time with osteogenesis; at D21, all the concentrations of GRA further increased the expression of BSP, and the concentration of 33 μM showed a statistically significant upregulation of BSP. A similar pattern was observed for the expression of SMAD1, where GRA induced a significant increase in gene expression only at the late time point D21 at the concentrations of 3.3, 10, and 100 μM. GRA was also shown to significantly affect the expression of CBS, one of the key enzymes implicated in the endogenous production of H_2_S, at day 21. A trend towards upregulation of gene expression after GRA stimulation was also observed for WNT16 at day 21, but due to high variability among different samples, it did not reach statistical significance. Occasional downregulations were observed upon GRA stimulation in the expression of ALP (at both D14 and D21) and WISP1 (at D14). It should be noted that both ALP and WISP-1 showed a strong upregulation in gene expression in unstimulated samples at D14 in this model.

## 4. Discussion

The steady increase in the ageing population and the high prevalence of bone fragility makes studies on complementary therapies and the use of functional food and phytocompounds a compelling field of interest for the prevention of bone loss and fracture. Epidemiological evidence strongly suggests that diets rich in fruit and vegetables can prevent or delay the onset of a wide range of noncommunicable diseases [47]; musculoskeletal health is no exception to this notion [1].

Phytochemicals belonging to the Brassicaceae family have drawn particular interest in past decades. They are rich in organosulfur compounds, which exert a strong antioxidant and cytoprotective activity, owing to their fundamental role in the defence system of Brassicaceae against exogenous stress [48]. Clinical studies demonstrated that a high intake of cruciferous vegetables for a period of 14.5 years is associated with a lower hazard of fracture and a lower incidence of injurious falls-related hospitalization in a cohort of postmenopausal women, suggesting that GLS and ITC may play an important role in stimulating bone metabolism [4,49]. 

In this work, we showed that GRA, a GLS abundant in several plants belonging to the Brassica species, induces osteogenic differentiation of hMSCs while increasing H_2_S intracellular levels. 

The capacity to release H_2_S has been previously recognized as one important mechanism by which naturally occurring polysulfides and ITCs acts on cells and tissues [50]. Particularly, Benavides et al. first demonstrated that diallyl disulfide (DADS) and triallyl trisulfide (DATS), two of the main polysulfides obtained from the catabolic breakdown of allicin in garlic, release H_2_S via thiol-dependent mechanisms and that H_2_S mediated the vasoactivity of garlic [50]. Recently, a similar chemical and biological mechanism was shown to account for H_2_S release and vasorelaxant effect of ITC [51], leading to hypothesize that the capacity to release H_2_S, which is a common feature in natural ITCs, is the key determinant of the multiple biological effects of these organosulfur compounds [40]. Following a previous report [24], we confirmed that GRA can release H_2_S in solution in a cysteine-dependent manner. Given that this system does not contain myrosinase or myrosinase-like activity, this finding is suggestive of an intrinsic capacity of GRA to release H_2_S independent of its conversion to SFN. That a prototypical GLS such as GRA may release H_2_S is not an obvious finding, and the detailed chemical mechanism underlying this evidence has yet to be elucidated. Consequently, in this work, we undertook a number of assays to ascertain, on the one hand, that the detection of H_2_S was technically accurate and, on the other hand, to gain insights on whether the increase in intracellular H_2_S level observed after stimulation with GRA is biologically relevant in hMSCs. Our findings showed that GRA induces a detectable increase in the amount of intracellular H_2_S in a dose-dependent fashion in hMSCs adherent to wells, and these findings were confirmed after detaching the cells and analyzing cell suspensions by flow cytometry, thereby excluding potential artefacts linked to any leaking of the H_2_S-selective probe in the extracellular environment. 

The evidence of direct bioactivity of GRA on cells and tissues is potentially relevant in humans. Despite a lower bioavailability and greater interindividual variation in excretion than their hydrolytic products ITCs, GLSs are entailed with longer half-life and slower elimination due to their greater chemical stability. Moreover, although most GRA introduced with diet undergoes hydrolysis in the gut, a fraction of GRA is absorbed directly in the stomach and the small intestine before the catabolic breakdown to SFN is triggered by gut microbiota [52,53]. It is conceivable that this unprocessed GRA may exert additional biological activity in tissues. Aligned with these hypotheses, we found that long-term stimulation with GRA increased the phenotypical maturation of hMSCs by stimulating mineral apposition and inducing a statistically significant increase in the expression of SMAD1, CBS, and BSP. Moreover, the upregulation of CBS, which is one of the enzymes responsible for H_2_S within mammalian cells, indicates positive feedback of the regulation of H_2_S by GRA. 

As the hydrolytic product of GRA, SFN had been already shown to stimulate bone formation and inhibit the activity of osteoclast in bone [54,55]. It can be suggested that the ‘GRA–SFN system’ exerts a beneficial effect on bone both at the level of GLS and of its cognate ITC [56]. Moreover, these data further confirm previous evidence from our group and others that the capacity to release H_2_S is linked to the stimulation of osteogenesis both in vitro and in vivo [35,36,38,44]. Therefore, findings from this work extend the notion that H_2_S-donors play an anabolic role in the bone to dietary organosulfur compounds and may constitute a likely biological mechanism for the improved skeletal health observed in patients who maintain elevated intake of cruciferous vegetables [4,49]. 

Whether these properties are also maintained by other GLS family members is unknown and may constitute an interesting further development of this study. In this context, it should be noted that one recent study demonstrated that the ethanol extract of Brassica rapa (turnip), containing 13 GLSs belonging to the three distinct chemical GLS subsets (aliphatic, aromatic, and indole), induced a stimulatory effect on bone formation both in vitro and in vivo [57], but whether this effect is linked to the release of H_2_S in unclear. 

These findings, which highlight a direct effect of unprocessed GRA on hMSCs derived from bone tissue, may indicate that GRA administration can be beneficial for bone tissue even without consumption of myrosinase-containing powder [56], thus reviving the interest in studies devoted to increasing quantities of bioactive GRA in foods either by postharvest strategies [58] or metabolic engineering in plants [59] and microbial source [60]. 

Overall, these findings lay the ground for further studies on nutraceutical-based complementary medicine in bone health. Particularly, studies aimed to correlate nutrient intake, H_2_S blood levels, and bone status would help define preventive/clinical dietary protocols for patients with an increased risk of bone fragility.

## 5. Conclusions

Our data confirmed an H_2_S release by GRA in a buffer in the presence of L-cysteine and first showed that hMSCs exposed to GRA increased intracellular H_2_S levels. Furthermore, the myrosinase-free in vitro model used in this study helped to confirm a biological property of GRA independent of the catabolic hydrolysis to generate active SFN. Specifically, our data showed stimulation of osteogenic differentiation similar to those obtained by chemical and pharmacological H_2_S donors. In particular, GRA upregulated genes correlated to osteogenic differentiation and increased mineral apposition. This study increases the body of evidence showing that GLSs not only work as precursors of their reactive derivatives ITC but exert biological properties mediated, at least in part, by the release of H_2_S. Notably, our study offered the first evidence of a biological effect of GRA in human bone cells, particularly in the stimulation of osteogenic differentiation. In doing so, it laid the ground for further studying the use of cruciferous derivatives as natural alternatives to chemical H_2_S donors as adjuvant therapies in treating bone-deteriorating pathologies.

## Figures and Tables

**Figure 1 nutrients-14-00435-f001:**
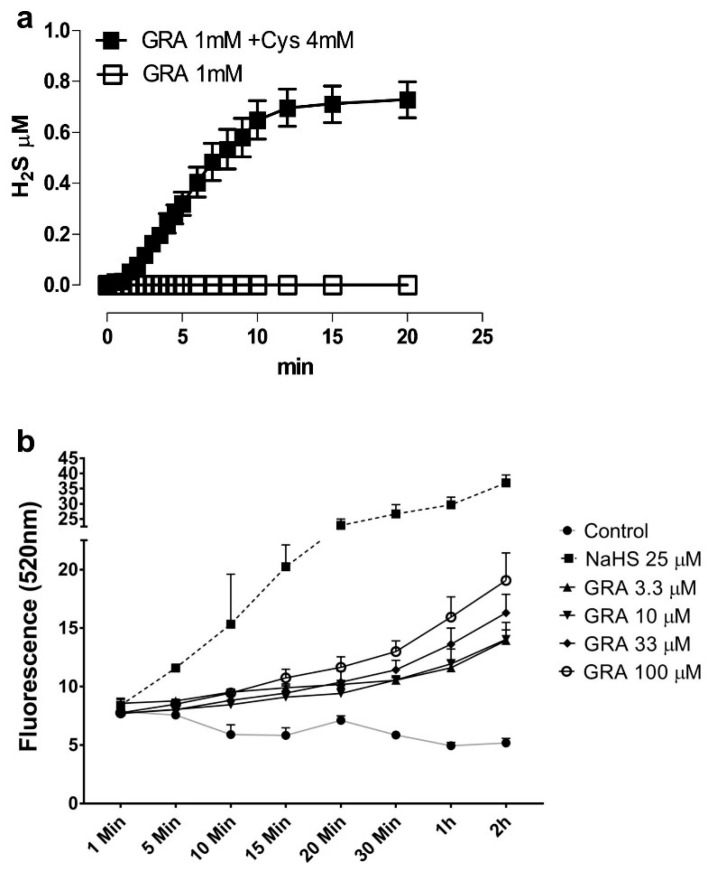
Detection of H_2_S levels in GRA dissolved in aqueous buffer and time course detection of H_2_S levels in hMSCs after stimulation with GRA. (**a**) Graphs of H_2_S concentration in the buffer as detected by amperometric measurements (*N* = 3 independent experiments). (**b**) Graphs of time-course measurement of intracellular H_2_S in hMSCs stimulated with 3.3–100 μM concentrations of GRA at the indicated time points as detected by fluorimetric analysis. Cells treated with 25 μM NaHS were used as a positive control; cells untreated were used as negative controls (*N* = 3 independent experiments).

**Figure 2 nutrients-14-00435-f002:**
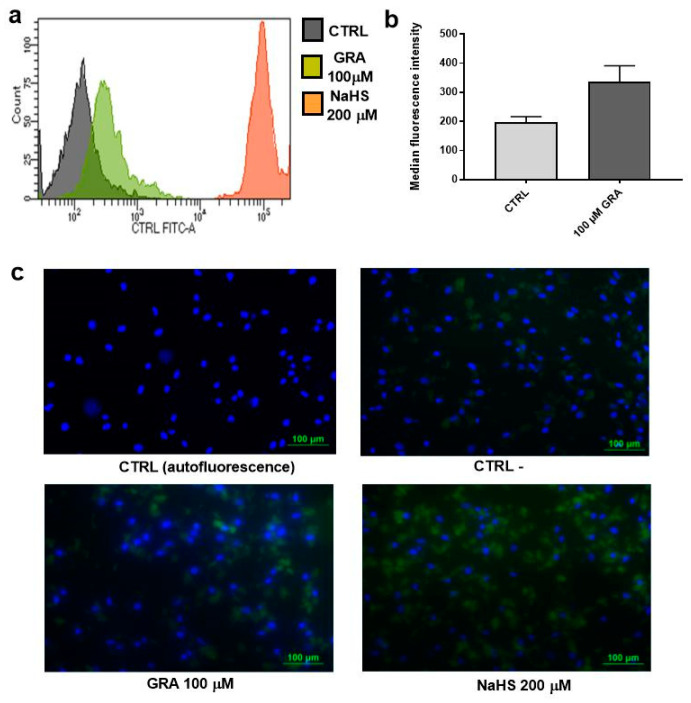
Measurements of intracellular H_2_S levels in hMSCs after stimulation with GRA. (**a**) Histograms showing representative peak fluorescence intensities in GRA-treated hMSCs. Cells treated with 200 μM NaHS were used as a positive control. (**b**) Histograms showing the median fluorescence intensity in GRA-treated hMSCs vs. CTRL hMSCs (*N* = 3 independent experiments). Wilcoxon signed-rank test was performed for the statistical analysis. (**c**) Representative immunofluorescence pictures showing the intracellular H_2_S staining (CTRL = cells without WSP-5; CTRL- = cells labelled with the probe WSP-5, without treatment). Magnification 100×.

**Figure 3 nutrients-14-00435-f003:**
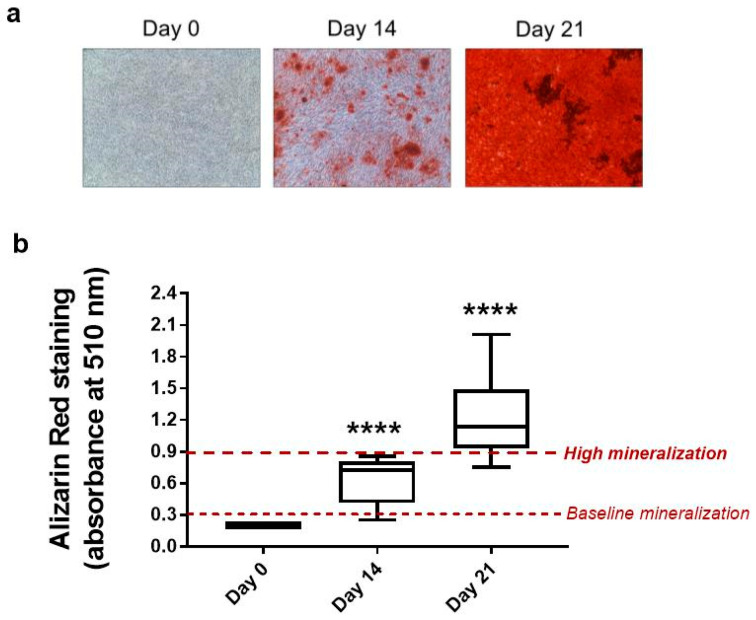
Mineral apposition in control osteogenic cultures of hMSCs. (**a**) Representative pictures of Alizarin Red staining in osteogenic cultures of hMSCs (magnification 100×). (**b**) Box plot showing the quantification of mineral apposition at day 0, day 14, and day 21 in control cells. Data are expressed as median ± min–max values of six independent experiments. Wilcoxon signed-rank test vs. 0.2 value (no mineralization) was performed for the statistical analysis (**** *p* < 0.0001 vs. control day 0).

**Figure 4 nutrients-14-00435-f004:**
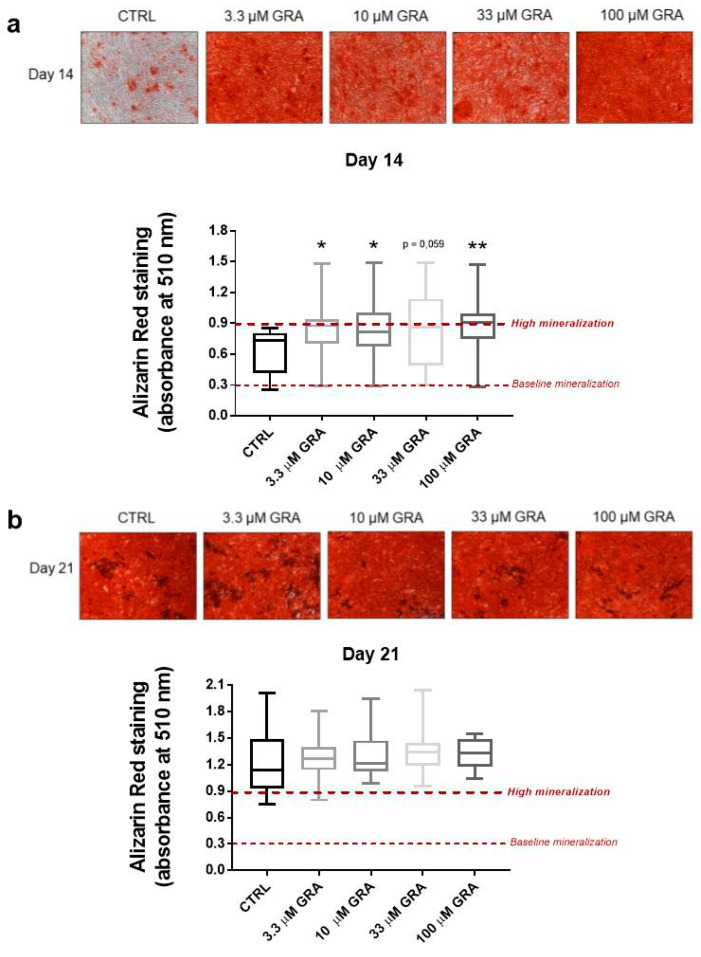
Mineral apposition in hMSCs after stimulation with GRA; Representative pictures (magnification 100×) and box plot showing the quantification of mineral apposition at day 14 (**a**) and day 21 (**b**) in hMSCs stimulated with 3.3–100 μM concentration of GRA (*N* = 6 independent experiments, showed as median, min—max values). One-way ANOVA and Dunnett’s multiple comparison tests were performed for the statistical analysis (* *p* < 0.05 and ** *p* < 0.01 vs. control day 14).

**Figure 5 nutrients-14-00435-f005:**
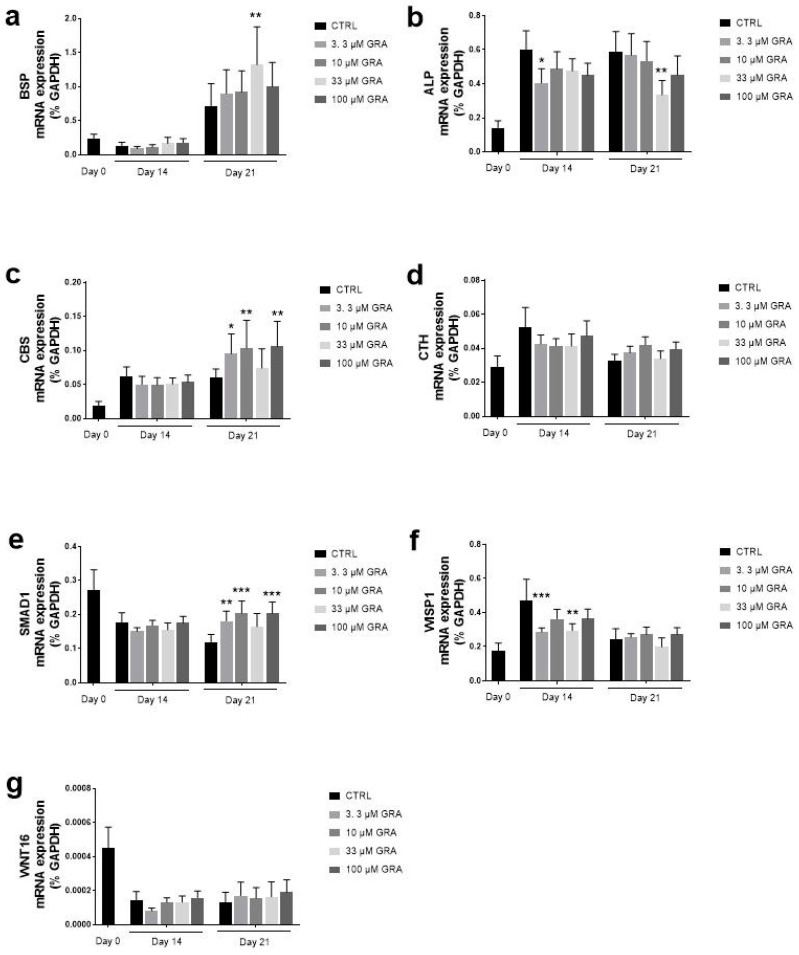
mRNA expression of osteogenic markers in hMSCs after stimulation with GRA. Histograms of mRNA expression of BSP (**a**), ALP (**b**), CBS (**c**), CTH (**d**), SMAD 1 (**e**), WISP-1 (**f**), WNT16 (**g**) at day 0, day 14, and day 21 and are expressed as mean ± SEM of six independent experiments. Two-way ANOVA with repeated measures by two factors (time and treatment) and Dunnett’s multiple comparison test vs. control cells at each time point were performed for statistical analysis (* *p* < 0.05, ** *p* < 0.01, *** *p* < 0.001 vs. control at day 14 or day 21).

**Table 1 nutrients-14-00435-t001:** List of primers sequences. FW: forward primer; REV: reverse primer.

Gene	Protein		5′-Sequence-3′	Product Size (bp)	Accession Number
GAPDH	Glyceraldehyde-3 phosphate dehydrogenase	FW	CGGAGTCAACGGATTTGG	218	NM_002046
		REV	CCTGGAAGATGGTGATGG		
CBS	Cystathionine-β-synthase	FW	AATGGTGACGCTTGGGAA	107	NM_000071
		REV	TGAGGCGGATCTGTTTGA		
CTH	Cystathionine-γ-lyase	FW	AAGACGCCTCCTCACAAGGT	170	NM_001902
		REV	ATATTCAAAACCCGAGTGCTGG		
ALP	Alkaline phosphatase	FW	GGAAGACACTCTGACCGT	152	NM_000478
		REV	GCC CAT TGC CAT ACA GGA		
BSP	Bone sialoprotein	FW	CAGTAGTGACTCATCCGAAG	158	NM_004967
		REV	CATAGCCCAGTGTTGTAGCA		
WNT16	Wnt Family Member 16	FW	GCCAGTTCAGACACGAGAGA	140	NM_057168
		REV	TGCAGCCATCACAGCATAAA		
SMAD1	SMAD Family Member 1	FW	CACCCGTTTCCTCACTCTCC	257	NM_005900
		REV	TCCTCATAAGCAACCGCCTG		
WISP1	WNT1-inducible-signalling pathway protein 1	FW	ACACGCTCCTATCAACCCAAG	103	NM_003882
		REV	CATCAGGACACTGGAAGGACA		

## Data Availability

The data presented in this study are available on request from the corresponding author.

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
