# Peer review of "Glucoraphanin Increases Intracellular Hydrogen Sulfide (H2S) Levels and Stimulates Osteogenic Differentiation in Human Mesenchymal Stromal Cell"

_nutrients, 2022, doi:10.3390/nu14030435_

Round 1
Reviewer 1 Report
Gambari and colleagues have evaluated a glucosinolate, glucoraphanin obtained from Tuscan black kale, in osteogenic potential of human mesenchymal stromal cells. Since abnormal bone remodeling caused by skeletal aging or estrogen deficiency is a very important clinical issue, we need to better understand the role of natural compound in the pathologic condition. Here, the authors claim that H2S release by glucoraphanin stimulates osteoblast differentiation and increases mRNA levels of several known osteogenic markers including Wnt signaling. The manuscript is clearly written and the approaches to make their work-flows seem appropriate. However, as considering this journal’s impact, the authors have to assess any in vivo experiments to investigate whether glucoraphanin affects bone formation/resorption as well as health span at the same condition. In addition, they do not have to ignore the effect of H2S release by glucoraphanin on osteoclast differentiation or activation because H2S acts by inhibiting osteoclast differentiation, as the authors mentioned in the Introduction. More seriously, there are many statistic issues here in the manuscript. For example, this reviewer could not find any error bars in all the control groups even though the authors said ‘control samples’. It seems like they did not use replications for the controls. If this is the case, all the data presented in the manuscript should be reconsidered and verified with any statistic experts before submitting to the journal. In addition, the entire Figure 3 does not need to be shown because this is well established data by numerous research groups. The authors can mainly focus on the effect of H2S release by glucoraphanin on osteoblast/osteoclast differentiation in this study. Lastly, since they could not find any significant change in osteogenic potential (Fig.4 b and c), their conclusion should be changed or at least town down to explain the role of glucoraphanin in osteoblast differentiation of human mesenchymal stromal cells.
Specific points.
- Introduction: As the authors mentioned, glucosinolate is converted to sulforaphane, a potent inducer of the KEAP1/NRF2/ARE pathway, leading to the activation of a potent antioxidant and detoxifying response in cells as well as anti-inflammatory and anti-apoptotic effects. Did they check this signaling pathways in the presence of glucoraphanin?
- Material and Methods: The patients’ information is missing. Need to report on details about their sex, age, and number.
- Lane 162: ‘Each analysis was performed in duplicate’. If this is the case, how did they make all the statistics (error bars from experimental groups)?
- Lane 176: The authors need to explain what passage 3 means. It seems they used a different cell culture conditions compared to the osteogenic differentiation assay.
- Lane 234: Figure 2b should be replaced to Figure 2c.
- Lane 267: They cannot say ‘marked’.
- What is the rationale of Figure 5? The osteogenic potential was not significant in the presence of glucoraphanin.
Reviewer 2 Report
The results of this manuscript may be interesting but before publication some points need to be resolved as follows:
Lines 215 and 222: is the concentration of NaHS 25 mM correct? In figure 1b the concentration is 25 µM.
Why, in the experiments reported in Figure 2, NaHS was used at 200 µM instead of 25 µM?
Line 239: is the concentration of NaHS 200 mM correct? In figure 2a the concentration is 200 µM.
Figure 2b: how are the results expressed? What does the error bar represent? Why is there no error bar in the CTRL? Please perform statistical analysis to assess the significance of the effect
Figure 3: this figure has nothing to do with the GRA, please explain the meaning and usefulness of this figure.
Why in the figure 4a and 4b, in the CTRL is missing the SEM bar?
Figure 4: this figure is not convincing: the effect is small, has a very high SEM and is not dose-dependent. Why the SEM bar is missing in the CRTL?
Line 267: I think that the increase in the mineral matrix apposition is not marked but is small. Considering also the very high SEM values reported in figure 4, it is difficult to believe that the effect is statistically significant. In particular, at 21 days there is no effect (when the statistical analysis shows the absence of significance, it means that the observed effect is due to chance and not to treatment).
Figure 5: the same considerations reported for figure 4 are valid for this figure
Round 2
Reviewer 1 Report
The patients' information is still missing in the Method section. Beside of that, the authors have well-addressed the critiques from this reviewer and the manuscript has been improved much better than before.
Reviewer 2 Report
The authors have responded comprehensively to all criticisms, the manuscript has been greatly improved and can be published without further modification.